# Mass Spectrometric Identification of *Licania rigida* Benth Leaf Extracts and Evaluation of Their Therapeutic Effects on Lipopolysaccharide-Induced Inflammatory Response

**DOI:** 10.3390/molecules27196291

**Published:** 2022-09-23

**Authors:** Thayse Evellyn Silva do Nascimento, Jorge A. López, Eder Alves Barbosa, Marcela Abbott Galvão Ururahy, Adriana da Silva Brito, Gabriel Araujo-Silva, Jefferson Romáryo Duarte da Luz, Maria das Graças Almeida

**Affiliations:** 1Post-Graduation Program in Pharmaceutical Sciences, Health Sciences Center, Federal University of Rio Grande do Norte, R. Gen. Gustavo Cordeiro de Farias, Petrópolis 59012-570, Natal/RN, Brazil; 2Multidisciplinary Research Laboratory, DACT, Health Sciences Center, Federal University of Rio Grande do Norte, R. Gen. Gustavo Cordeiro de Farias, Petrópolis 59012-570, Natal/RN, Brazil; 3Study Group on Bioprospecting and Oxidative Stress of Natural Products, Amapá State University (UEAP), Av. Presidente Vargas, Centro 68900-070, Macapá/AP, Brazil; 4Laboratory of Synthesis and Analysis of Biomolecules (LSAB), Institute of Chemistry, Darcy Ribeiro University Campus, University of Brasilia, Brasilia 70910-900, Brasília/DF, Brazil; 5Faculty of Health Sciences of Trairi (FACISA/UFRN), R. Passos de Miranda, Santa Cruz 59200-000, Santa Cruz/RN, Brazil; 6Organic Chemistry and Biochemistry Laboratory, Amapá State University (UEAP), Av. Presidente Vargas, Centro 68900-070, Macapá/AP, Brazil

**Keywords:** plant extract, phytocomposition, chromatography, anti-inflammatory

## Abstract

*Licania rigida* Benth has been evaluated as an alternative drug to treat diseases associated with inflammatory processes. This study evaluated the anti-inflammatory effects of aqueous and hydroalcoholic leaf extracts of *L. rigida* with inflammation induced by lipopolysaccharides in in vitro and in vivo inflammation models. The phytochemical profile of the extracts, analyzed by ultra-fast liquid chromatography coupled with tandem mass spectrometry, revealed the presence of gallic and ellagic acids in both extracts, whereas isovitexin, ferulate, bulky amino acids (e.g., phenylalanine), pheophorbide, lactic acid, and pyridoxine were detected in the hydroalcoholic extract. The extracts displayed the ability to modulate in vitro and in vivo inflammatory responses, reducing approximately 50% of pro-inflammatory cytokine secretion (TNF-α, IL-1β, and IL-6), and inhibiting both NO production and leukocyte migration by approximately 30 and 40% at 100 and 500 µg/mL, respectively. Overall, the results highlight and identify, for the first time, the ability of *L. rigida* leaf extract to modulate inflammatory processes. These data suggest that the leaf extracts of this plant have potential in the development of herbal formulations for the treatment of inflammation.

## 1. Introduction

The inflammatory process involves a complex series of tightly controlled biochemical cellular events that evolve to contain or eliminate foreign infectious agents and repair tissue damage. This response is normally beneficial and is necessary for the organism to self-regulate and quickly restore homeostasis. An inefficient or uncontrolled immune response promotes cellular dysfunction, tissue damage, and inadequate repair, which are characteristics of many inflammatory diseases [1,2].

Although important for the human body, these responses must be efficiently regulated to prevent the development and worsening of inflammatory diseases. Therefore, several cellular mediators are secreted that perform essential functions to achieve homeostasis. White blood cell infiltration is pivotal for these inflammatory processes [3,4].

Macrophages are dynamic monocyte-derived cells that play a critical role in immune cells by activating inflammatory pathways and releasing inflammatory mediators [5]. Macrophages provide three main functions in relation to inflammation: antigen presentation, phagocytosis, and immunomodulation through the mediation of several cytokines and growth factors. Macrophages are closely related to the triggering, maintenance, and termination of inflammation. Therefore, these cells are considered one of the main cytokine sources involved in the immune system, and their activation promotes the synthesis and release of several cytokines, such as TGF-ß1, TNFα, TNF- γ, IL-6, IL-8, IL-10, IL-1 γ, PGE-2, nitric oxide, and rG-CSF. IL-8 and IL-6 are potent inflammatory cytokines, whereas IL-10 and TGF-ß1 are important anti-inflammatory cytokines that exhibit the ability to deactivate macrophages [1,6,7,8].

An inefficient or decompensated response contributes to the cellular dysfunction, tissue damage, and inadequate repair that are present in many inflammatory diseases. Thus, during an exacerbated response, anti-inflammatory drugs can mitigate deleterious effects on the human body. Hence, non-steroidal anti-inflammatory drugs (NSAIDs) are clinically administered, although their prolonged use causes serious side effects, such as iron deficiency anemia, gastric ulcers, liver and kidney toxicity, and gastrointestinal bleeding, with a concomitant increase in morbidity and mortality rates. These negative effects are a cause for concern, as anti-inflammatory drugs are used indiscriminately worldwide by individuals of all age groups. Therefore, studies have focused on natural compounds as alternative treatments to modulate the inflammatory response, particularly molecules with relatively few side effects, especially for long-term use [4,9]. Medicinal plants are a reservoir of chemical substances whose therapeutic properties in the human body must be carefully analyzed. Many of these substances, known as active principles, are transformed into drugs suitable for treating various human diseases [10,11].

Brazilian biodiversity is prevalent worldwide, with approximately 46,000 catalogued species. The vegetation of the Caatinga biome in Brazil is poorly researched, and studies are required to ensure the safe use of plant species to which folk medicine attributes pharmacological properties. *Licania rigida* Benth is a large evergreen tree species from the Brazilian Caatinga that is known as oiticica, and is commonly used in the treatment of inflammatory processes and diabetes [12]. This plant is traditionally used for its antimicrobial and anticancer properties, which are associated with oxidative stress [13,14,15,16]. Studies have evaluated the biological and pharmacological activities of plants belonging to the same family as *Licania* (Chrysobalanaceae) and demonstrated their anti-inflammatory effects [13,17].

Several studies have reported the use of extracts from different plant parts of the Chrysobalanaceae family, such as the bark, leaves, and fruit, for therapeutic and prophylactic purposes, including the treatment of inflammatory processes, whereby a decrease in pro-inflammatory cytokine expression was observed [13,18,19,20,21]. A hydroalcoholic leaf extract of *L. rigida* was evaluated in a mouse paw edema model, where acute inflammation was induced by carrageenan. The peritoneal exudate was assessed for vascular permeability and leukocyte migration, and an anti-inflammatory effect was observed, mainly related to the inhibition of monocyte and neutrophil migration [20].

Several plant metabolites of *L. rigida* have shown pharmacological activity, including certain phenolic compounds. According to Morais et al. [16], *L. rigida* crude leaf extract and its ethyl acetate fraction contain the flavonol-*O*-3 glycosylated flavonoid as its main compound. This molecule displays several biological properties, including anti-inflammatory activity and inhibitory potential against inflammatory mediators such as xanthine oxidase (XO), cyclooxygenase (COX), and lipoxygenase [22,23]. *L. rigida* also shows anticoagulant activity, as its alcoholic leaf extract directly inhibits thrombin [24]. Pro-inflammatory cytokines are closely related to the thromboembolic process by modulating coagulation activation through thrombin production and attenuation of endogenous fibrinolysis. Therefore, inflammation is strongly linked to the pathophysiology of venous thromboembolism [25,26,27].

Despite the use of medicinal plants in the treatment and prevention of diseases due to their pharmacological properties, there remain concerns about the toxicity, cytotoxicity, genotoxicity, and mutagenicity of these compounds. Many plant species have toxic constituents that are responsible for triggering hepato- and renal toxic effects, abortion, and poisoning [28,29,30,31]. However, Luz et al. [24] and Batista et al. [32] showed no toxic, cytotoxic, or genotoxic effects when using alcoholic and aqueous leaf extracts of *L. rigida* in vivo or in vitro. Therefore, the use of these extracts is promising and appear safe from a toxicological perspective.

Based on these considerations, *L. rigida* shows promising pharmacological activities, as described in the literature. Nonetheless, a more thorough analysis of these activities is required due to the indiscriminate use of this plant in folk medicine and the urgent need for alternatives to anti-inflammatory therapy, considering the undesirable reactions resulting from conventional NSAID treatment. Studies on the anti-inflammatory potential of plant species would reveal their phytocomposition and indicate potential pharmacological applications. Hence, this study analyzed the chemical composition of *L. rigida* leaf extracts and evaluated their anti-inflammatory effects by applying an in vitro model using LPS-stimulated RAW 264.7 macrophages and an in vivo model of LPS-induced peritonitis to contribute to the prospection of new anti-inflammatory molecules with low side effects.

## 2. Results

Aqueous and hydroethanolic leaf extracts of *L. rigida* were analyzed by LC–MS/MS, and their spectra were compared with those of the GNPS database to identify the detected compounds. Despite the high number of MS/MS spectra acquired for each extract, only those in the database that matched with a cosine ≥0.85 and a mass difference ≤0.005 were considered for analysis. The chemical profiles of both extracts showed the presence of gallic acid, a metabolite of pharmacological interest, as well as ellagic acid. Other constituents included adenosine monophosphate, phenylalanine, vitamin B6 (pyridoxine), and isovitexin. Furthermore, the antioxidants ferulic acid and pheophorbide A and a lactic acid derivative were identified in the hydroalcoholic extract.

The extracted ion chromatograms (XICs) obtained for each structure and identified by UPLC-MS/MS and GNPS analyses showed four main phytocomponents with clear resolution for the *L. rigida* aqueous extracts (Figure 1A) and nine structures in the hydroethanolic extracts (Figure 1B). Table 1 shows the phytochemical components of both the extracts and their respective cosines, mass differences, masses, molecular formula, ion fragments, and adducts resulting from the ionization process.

After phytochemical characterization of both *L. rigida* leaf extracts, their respective cytotoxicities were evaluated on activated macrophages (RAW 264.7). Results from cell viability assays using MTT (Figure 2A) and Alamar Blue^®^ (Figure 2B) showed no cytotoxic effects after challenging activated macrophages with different extract concentrations, compared with the control group.

Leukocyte migration in the inflammatory process is of paramount importance because it is responsible for the induction, maintenance, and regulation of immune responses [33]. The AELR and HELR extracts significantly decreased leukocyte expression after treatment, both in terms of total and differential leukocyte counts (Figure 3A,B).

LPS is recognized by the toll-like receptor (TLR4), which is part of the macrophage constitution and activates these defense cells in the presence of LPS. Consequently, several signaling pathways are activated to release pro-inflammatory mediators, such as tumor necrosis factor (TNF-α), interleukin-1 beta (IL-1β), interleukin-6 (IL-6), and nitric oxide (NO), to achieve homeostasis [34]. After AELR and HELR extract administration, the results showed a marked reduction in inflammatory cytokine secretion (TNF-α and IL-1β) in activated RAW 264.7 macrophages by LPS (Figure 4 and Figure 5).

Regarding the inhibition of TNF-α, the best result was observed at the highest concentrations of both extracts, characterizing a concentration-dependent response (Figure 4). HELR displayed the most efficient reduction in TNF-α compared with AELR at the highest extract concentration, whereas the latter showed a better result than HELR at a concentration of 100 µg/mL (Figure 4A,B). Statistical analysis between both extracts determined a value of *p* = 0.023 (HELR over AELR) and *p* = 0.045 (AELR over HELR), respectively, using a one-way ANOVA followed by Tukey’s post hoc test.

Both extracts displayed a slightly significant reduction in IL-1β cytokine expression compared with the positive control. Nevertheless, an increase in the extract concentration significantly reduced IL-1β expression (Figure 5). HELR at 500 µg/mL exhibited superior results (Figure 5B), which were significantly different from those of AELR at the same concentration (Figure 5A). The statistics between the extracts had a *p* value = 0.039 (HELR over AELR), using a one-way ANOVA followed by a post hoc Tukey test.

AELR was less effective in modulating IL-6 cytokine secretion, reaching a reduction of approximately 50% at an extract concentration of 500 µg/mL (Figure 6A), whereas HELR showed a more satisfactory inhibition of IL-6 expression as extract concentration increased, achieving its highest capacity at 500 µg/mL (Figure 6B). The statistics between the extracts determined a *p* value = 0.042 (HELR over AELR), using a one-way ANOVA followed by a post hoc Tukey test.

Acute inflammation was LPS-induced in vivo in the peritoneal cavity of C5,7BL/6 male mice to determine the ability of the *L. rigida* extract to inhibit inflammatory cytokine infiltration at the injury site. The LPS-stimulated animals (positive control) showed a significant increase in inflammatory cytokines in the peritoneal cavity (*p* < 0.05) compared with the non-stimulated group (negative control), confirming the inflammatory process induced by LPS (Figure 7A–C). In animals with LPS-induced inflammation, treatment with aqueous and hydroethanolic extracts resulted in a significant decrease in inflammatory cytokine infiltration into the peritoneal cavity (*p* < 0.05). Both the aqueous and hydroethanolic extracts inhibited TNF-α secretion, although HELR displayed a more satisfactory result, with a reduction of approximately 50% (Figure 7A). Both extracts reduced IL-1β secretion, with HELR reaching greater than 50% inhibition (Figure 7B). Both extracts reduced IL-6 secretion by approximately 50% (Figure 7C).

Both the aqueous and hydroethanolic extracts promoted a significant reduction in NO production in the macrophages at 100 and 500 µg/mL (*p* < 0.05) compared with the LPS-stimulated control group (Figure 8A,B). AELR and HELR displayed similar results, inhibiting NO secretion by approximately 30 and 40% at 100 µg/mL and 500 µg/mL, respectively. No significant differences were observed between the negative controls. With respect to in vivo NO secretion, local inflammation was LPS-stimulated in the peritoneal cavity of male mice. Animals stimulated with LPS (positive control) showed a significant increase in NO in the peritoneal cavity (*p* < 0.05) compared with non-stimulated animals (negative control), confirming the development of the inflammatory process (Figure 9). However, after AELR and HELR treatment, a reduction in NO production by approximately 50% was observed in the peritoneal cavity of LPS-stimulated animals.

## 3. Discussion

NSAIDs are widely used clinically as anti-inflammatory drugs, despite triggering undesirable adverse effects, such as gastrointestinal bleeding. These drugs are used to treat intestinal inflammation, including irritable bowel syndrome (IBS), which corresponds to a group of chronic inflammatory diseases such as Crohn’s disease and ulcerative colitis, and NSAIDs can exacerbate these pathologies. Moreover, chronic use of this class of drugs is responsible for the development of these pathologies, and patients often use several drugs to treat inflammation. Currently, the situation is aggravated due to the unavailability of effective drugs with few side effects for management of these diseases. Hence, studies have demonstrated the effectiveness of natural and herbal products for the treatment of Crohn’s disease and ulcerative colitis [34,35,36].

NSAIDs also cause hypersensitivity reactions in patients, which can result in anaphylaxis and death [37]. The indiscriminate use of NSAIDs has stimulated the search for new therapeutic methods and medicinal plants represent a reservoir of chemical compounds that show strong potential in the development and production of new and effective drugs [38].

The bioactivities attributed to phytochemical compounds have generated scientific interest for further study of their therapeutic properties, including antioxidant and anti-inflammatory effects [39]. Hence, this study qualitatively analyzed the phytocomposition of aqueous and hydroethanolic leaf extracts of *L. rigida* using mass spectrometry and evaluated their anti-inflammatory and cytotoxic effects in in vitro and in vivo models. Although phytochemical quantification is required to validate the relevance of these extracts, the phytocomposition of the genus *Licania* has previously been characterized [13,14,21].

No cytotoxic effects of AELR and HELR were observed on cell viability using the Alamar Blue and MTT assays. Furthermore, previous studies by our group with AELR indicated no toxic effects, as evidenced by acute toxicity, 28-day repeated dose oral toxicity, cytotoxicity, and mutagenicity [32]. Regarding the hydroethanolic leaf extract, Morais et al. [16] reported no toxicity in animals, no cytotoxicity, and nonsignificant apoptotic levels. This assessment is required to determine the potential health risks of natural products [40].

Studies on the phytochemical characterization of *Licania* are scarce in the scientific literature. However, Carnevale et al. [21] reported that triterpenoids, diterpenoids, steroids, and flavonoids were the main chemical compounds in the Chrysobalanaceae family. Morais et al. [16] detected significant amounts of phenolic compounds and flavonoids with flavonol-3-*O*-glycosylates as the main constituents in the phytochemical analysis of *L. rigida* hydroalcoholic leaf extract and its ethyl acetate fraction. This flavonol is likely isovitexin, which was identified in the hydroalcoholic extract in the present study, although further analyses, such as NRM, are required to confirm this finding. Moreover, other studies have analyzed different extracts and fractions of *L. rigida* leaves and seeds and identified catechins, chalcones, flavonoids, and tannins in their chemical profiles [13,19]. Our AELR and HELR phytochemical analyses also identified compound classes similar to those described in these studies.

The anti-inflammatory effects of AELR and HELR may be explained by their flavonoid content. This class of metabolites displays anti-inflammatory and immunomodulatory properties [41,42], as evidenced by the decrease in inflammatory cytokines IL-6, TNF-α, and IL-1β, after *L. rigida* extract treatment. This result is consistent with scientific evidence that has shown that bioactive constituents, such as flavonoids and carotenoids in fruits, leaves, bark, and seeds, can play important anti-inflammatory roles, either individually or synergistically [43,44,45]. In addition, studies by Linus et al. [18] that used the bark of *Parinari kerstingii* (Chrysobalanaceae) highlighted aspirin-like results in inflammatory treatment of a carrageenan-induced paw edema model in Sprague-Dawley rats. Similarly, Venancio et al. [19] observed anti-inflammatory effects of *Chrysobalanus icaco* in decreased expression of pro-inflammatory cytokines, such as IL-1β and TNF-α.

Furthermore, the observed anti-inflammatory effect may be associated with isovitexin, which was identified in the *L. rigida* hydroethanolic leaf extract and has been shown to exhibit anti-inflammatory effects through the inhibition of several cytokines, such as IL-1β, IL-6, and TNF-α [46]. The AELR and HELR results are consistent with reports showing the anti-inflammatory potential of isovitexin in inhibiting cisplatin-induced renal injury in mice [47] and acute gouty arthritis in rats [48]. In both cases, a reduction in the expression and infiltration of the inflammatory cytokines IL-6, TNF-α, and IL-1β was observed after treatment with isovitexin.

This anti-inflammatory effect may also be associated with the presence of gallic acid in both AELR and HELR extracts. It is noteworthy that our results with both *L. rigida* extracts are corroborated by other reports, indicating that gallic acid is a subject of study due to its beneficial therapeutic activities, such as its role in the regulation of extracellular platelet-activating factor and modulation of the inflammatory process [49,50].

The identification of ellagic acid in the chemical composition of both *L. rigida* aqueous and hydroalcoholic extracts may also explain their remarkable effects in reducing inflammatory factors, as this compound is important because of its anti-inflammatory properties [51]. Data after treatment with AELR and HELR suggested the effective involvement of ellagic acid against LPS-induced inflammation. This result is consistent with the anti-inflammatory effects evidenced by Allam et al. [52] after treatment with ellagic acid in an animal model, showing a significant reduction in pro-inflammatory cytokine serum levels and a concomitant increase in anti-inflammatory cytokines. Studies have demonstrated that the therapeutic effects of ferulate are due to its antioxidant and anti-inflammatory properties [53], which may also be associated with the anti-inflammatory effects observed after treatment with *L. rigida* extracts using the LPS-induced inflammation model.

Adenosine monophosphate was also identified in the phytochemical profiles of the *L. rigida* extracts. This metabolite may be associated with anti-inflammatory effects following treatment with AELR and HELR. This compound is crucial for regulating inflammatory responses [54], as it activates a signaling pathway by suppressing or modulating IL-1β [55,56].

The in vivo anti-inflammatory effect of *L. rigida* extract also displayed a decrease in leukocyte migration and pro-inflammatory cytokine inhibition in the peritoneal cavity. A similar result was reported by Santos et al. [20], who evaluated the anti-inflammatory activity of *L. rigida* hydroethanolic leaf extract in a systemic inflammation mouse model. In both cases, the observed anti-inflammatory effect was likely due to the polyphenol contents of the extracts.

The efficient anti-inflammatory effects observed after *L. rigida* extract treatment suggest a synergistic effect from the different compounds identified in AELR and HELR. Studies have shown that drug combination is a useful strategy, as synergism offers opportunities to improve treatment effectiveness [57,58]. In plants, this synergy occurs because extracts are a mixture of secondary metabolites that can interact with each other, resulting in robust control to treat diseases [43,45]. In this regard, herbal extracts have been reported as anti-inflammatory agents [45], indicating the combined effects of these phytocompounds or their synergistic interactions to ameliorate inflammatory processes [59,60,61].

In addition to synergism, the chemical compositions of AELR and HELR display concurrent antioxidant activity, which is attributed to the presence of polyphenols. This activity was previously described by Morais et al. [16] and Batista et al. [32], who showed high scavenging free radical DPPH capacities and reductions in lipid peroxidation by HELR and AELR, respectively. Overall, examination of these antioxidant parameters indicates potential protective effects. Antioxidant activity is involved in the prevention of damage caused by ROS [62]. Free radical production by biological and environmental effects causes a natural antioxidant imbalance that triggers several inflammatory processes involved in the progression of various diseases [63].

Overall, natural products contain a diversity of compounds that can interact with different targets. Furthermore, some components of this phytocomposition can function as additives or synergists to exert therapeutic effects in association with other bioactive co-actives [11,64]. Therefore, natural products, as a complex mixture of molecules, have attracted scientific interest, considering the potential synergistic therapeutic effects of their chemical compositions [10,38,65].

Experimental data reveal the pharmacological potential of *L. rigida*, as evidenced by the anti-inflammatory effects of both the aqueous and hydroethanolic extracts, which were efficient in reducing leukocyte migration, and modulating inflammatory cytokine expression. Therefore, the chemical and biological results suggest the potential for prospecting safe molecules and formulations for the therapeutic management of inflammatory processes.

## 4. Materials and Methods

### 4.1. Collection of Plant Material and Preparation of Extracts

*L. rigida* leaves were collected in Florânia, RN, Brazil, in April 2021, under the approval of the Authorization and Information System in Biodiversity (SisBio) and the National System for the Management of Genetic Heritage and Associated Traditional Knowledge (SisGen). The species was identified by Dr. Jomar Gomes Jardim at the Department of Botany and Zoology Herbarium, Federal University of Rio Grande do Norte, Natal, RN, Brazil, under registration number 0674/08.

After selection, leaves were cleaned and air-dried at 40 °C for 48 h. The material was ground to sizes of 0.5–1.0 mm and stored in amber containers before extract preparation. The powdered material (300 g) was then subjected to decoction (100 °C/10 min) with 1.5 L of water, filtration, and lyophilization to obtain *L. rigida* aqueous leaf extract (AELR). For the *L. rigida* hydroethanolic leaf extract, 300 g of powdered leaves was macerated with a 1.5 L ethanol: water solution (50:50, *v/v*) for four days at room temperature. The extracts were filtered, rotaevaporated, and lyophilized to secure the HELR.

### 4.2. Phytochemical Analysis by Ultrafast Liquid Chromatography Coupled with Mass Spectrometry (LC-MS/MS)

Prior to LC-MS/MS analyses, extract samples were resuspended in methanol (µg/mL), centrifuged (30 min/13,000 rpm), filtered through a 0.22 μm membrane, and stored at −20 °C. AELR and HELR were diluted in pure acetonitrile (mobile phase) 10× and 20×, respectively.

Sample analysis was performed using ultrafast liquid chromatography in a UPLC Eksigent UltraLC 110-XL liquid chromatograph (AB Sciex, Framingham, MA, USA) coupled to a Kinetex 2.6 µm C18 100 Å column (50 mm × 2.1 mm) and a 5600+ TripleT spectrometer (AB Sciex). After equilibrating the column with 5% acetonitrile/0.1% formic acid for 5 min, the sample (2 µL) was automatically injected to perform separation with a linear gradient of 5% acetonitrile/0.1% formic acid (5–95%) for 10 min at a flow of 0.4 mL/min, keeping the column temperature at 40 °C. The mass spectrometer was operated in the positive information-dependent acquisition (IDA) mode with a mass range from 100 to 1800 *m/z* and a source temperature of 650 °C. IDA data acquisition was configured to fragment ions from 100 to 1250 *m/z*, with a load ranging from 1 to 3, and intensity greater than 1000 counts. Further acquisition parameters included: period cycle time = 900 ms; pulser frequency = 15,392 kHz; accumulation time = 250.0 ms; curtain gas = 15,000; ion source gas 1 = 50,000; ion source gas 2 = 45,000; and ion spray voltage floating = 5500. In addition to the extracts, a blank control was used. Prior to starting and following every five analyses, the spectrometer was calibrated with the calibration solution (AB Sciex) to an accuracy of approximately 0.5 ppm (sodium iodide (2 µg/µL) and cesium iodide (50 ng/µL) in 50/50 2-propanol/water).

For data analysis, the acquisition files (WIFF) were converted to mzXML format using MSConvert software (ProteoWizard 3.0, ProteoWizard, Palo Alto, CA, USA) and analyzed on the: Global Natural Products Social Molecular Networking (GNPS; http://gnps.ucsd.edu, accessed on 16 August 2022) platform for analysis with the Molecular-Library Search-V2 (version release_14) tool. Data were filtered by removing peaks with ~17 Da, referring to Molecules 2022, 27, and 1084 and 13 of 17 of the *m/z* values of the precursors present in the MS/MS spectra, selecting only the top six peaks in the 50 Da window across the spectrum. Data were then grouped by MS-Cluster with tolerances to an original mass of 0.02 Da and an ion of MS/MS fragments of 0.1 Da to create consensus spectra. Moreover, consensus spectra with fewer than two spectra were eliminated prior to the GNPS spectral library analysis. Library spectra were filtered according to the input data. All correspondences between the network and library spectra were required to exhibit a score >0.85 and at least four matching correspondences. The cosine score indicates a normalized dot product, which is a mathematical measure of the spectral similarity between two fragmentation spectra. A cosine score of 1 characterizes identical spectra and a cosine score of 0 denotes no similarity. (See Appendix A)

### 4.3. Cell Culture and Animals

For in vitro assays, murine macrophages (RAW 264.7) were purchased from the American Type Culture Collection (ATCC, Rockville, MD, USA) (ATCC^®^ TIB-71™). The cells were grown at 37 °C in a humidified atmosphere with 5% CO_2_ in Dulbecco’s modified medium (DMEM), containing 10% fetal bovine serum, streptomycin (5000 mg/mL), and penicillin (5000 IU).

For the in vivo assays, C57BL/6 male mice (25–30 g) were obtained from the UFRN Health Sciences Center Vivarium, Natal-RN, Brazil. Animals were kept in collective cages (*n* = 5) under controlled lighting conditions (12 h light/dark cycles) and temperature (22 ± 2 °C), with food and water provided ad libitum. All the experiments were approved by the UFRN Ethics Committee on Animal Use (Protocol No. 254.021/2021).

### 4.4. Cell Viability and Cytotoxicity Assays

RAW 264.7 macrophages were individually challenged in triplicate at concentrations of 5, 50, 100, and 500 µg/mL of AELR and HELR to assess cytotoxicity using the MTT assay (3-(4,5-dimethylthiazol-2-yl)-2,5-diphenyltetrazolium bromide). Macrophages (1 × 10^5^ cells/well) were seeded in 96-well microplates and incubated for 24 h at 37 °C to promote adhesion. Next, 100 µL of MTT (5 mg/mL) was added to each well, and the plates were incubated (37 °C for 4 h). After removing the culture medium, DMSO (100 µL) was added to each well and cell viability was assessed at 570 nm using a microplate ELISA reader (Epoch-Biotek, Winooski, VT, USA). Cells grown in DMEM were used as negative controls.

The cytotoxic effects of AELR and HELR at 5, 50, 100, and 500 µg/mL on cell viability were also evaluated in triplicate using the Alamar Blue^®^ assay. Briefly, macrophages (1 × 10^5^ cells/well) were seeded in 96-well microplates and incubated for 24 h to promote adhesion. After exposure to the extracts (24 h at 37 °C), 10% Alamar Blue^®^, corresponding to the medium volume in each well, was added. The plate was again incubated (4 h at 37 °C and 5% CO_2_), and reduced Alamar Blue^®^ was monitored at 570 and 600 nm using a microplate ELISA reader (Epoch-Biotek). Cells grown in DMEM were used as negative controls.

### 4.5. Leukocyte Migration into Peritoneal Cavity and Cytokine Dosage

C57BL/6 male mice were divided into four groups (*n* = 5), as follows: Group 1, negative control, receiving only PBS solution; Group 2, positive control; and Groups 3 and 4, treated with 25 mg/kg AELR and HELR, respectively. Groups 2, 3, and 4 were stimulated intraperitoneally with 2 µg/mL LPS (*E. coli* O55:B5 strain) to induce acute inflammation. After 15 min, AELR and HELR (25 mg/kg) were administered intravenously to Groups 3 and 4, respectively. Four hours later, the mice were anesthetized with xylazine and ketamine (1:1) and euthanized, and the abdominal cavity was washed with 2 mL of 0.5% saline solution and 1 mM EDTA before collecting peritoneal fluids. After recovery, total cells were counted using a hemocytometer, and the differential polymorphonuclear leukocyte (PMN) count was determined in eosin- and hematoxylin-fixed cytospin preparations. Peritoneal fluid samples from each group were stored at −80 °C to analyze TNF-α, IL-1β, and IL-6 levels using an enzyme immunoassay kit (ELISA), as well as NO production.

### 4.6. Cytokine Measurement (TNF-α, IL1-β, and IL-6)

Raw 264.7 cells (1 × 10^5^ cells/well) were seeded and stimulated with 2 µg/mL of LPS dissolved in DMEM for 1 h. Cells were then challenged with 5, 50, 100, and 500 µg/mL of AELR and HELR. After 24 h, the supernatants were harvested to determine TNF-α, IL1-β, and IL-6 levels using an immunoenzymatic assay (ELISA) kit (eBioscience), following the manufacturer’s instructions. Analyses were performed in triplicate to determine the optical density (OD) at 450 nm using a microplate ELISA reader (Epoch-Biotek). LPS-stimulated and unstimulated non-extract-treated cells were used as positive and negative controls, respectively.

The collected peritoneal fluid TNF-α, IL1-β, and IL-6 levels from each group after LPS-induced inflammation were measured using an enzyme-linked immunosorbent assay (ELISA) kit (eBioscience), following the manufacturer’s instructions. The OD was measured in triplicate at 450 nm.

### 4.7. Measurement of Nitric Oxide (NO) Production

Raw 264.7 cells (1 × 10^5^ cells/well) were seeded and LPS-stimulated (2 µg/mL dissolved in DMEM) for 1 h. Cells were then treated with 5, 50, 100, and 500 µg/mL of AELR and HELR. After 24 h, supernatants were collected to assess the total NO concentration by adding Griess reagent to 40 µL of supernatant and monitoring the absorbance at 545 nm using a microplate ELISA reader (Epoch-Biotek). LPS-stimulated and unstimulated cells that were not treated with the extract were used as positive and negative controls, respectively.

Regarding LPS-induced inflammation in mice, peritoneal fluid collected from each group was used to determine NO production. The total NO concentration was assessed after the addition of Griess reagent to 100 µL of peritoneal fluid, and the absorbance was measured at 545 nm. All measurements were performed in triplicate using a microplate ELISA reader (Epoch-Biotek).

### 4.8. Statistical Analysis

Data are expressed as means ± SEM and were analyzed with one-way ANOVA and Tukey’s post hoc test, using GraphPad Prism version 6.0 Software for Windows (GraphPad Software, San Diego, CA, USA). Statistical significance was set at *p* < 0.05.

## 5. Conclusions

The present study conducted a phytochemical analysis of *L. rigida* aqueous and hydroethanolic leaf extracts by LC-MS/MS, which showed a rich composition of phenolic compounds as well as flavonoids. Furthermore, these extracts promoted a significant anti-inflammatory effect in an in vitro model using murine macrophages activated with LPS and in an in vivo model of LPS-induced peritonitis. AELR and HELR caused a marked reduction in leukocyte migration to the mouse peritoneal cavity, in addition to reductions in the expression of inflammatory cytokines both in vivo and in vitro. *L. rigida* extracts also inhibited NO production in both experimental models. These results suggest that these extracts are associated with the inhibition of cytokine production, and the extract phytocomposition may be responsible for the observed anti-inflammatory activity. Although further studies are required, the data provide promising evidence supporting AELR and HELR as alternatives for prospective anti-inflammatory agents.

## Figures and Tables

**Figure 1 molecules-27-06291-f001:**
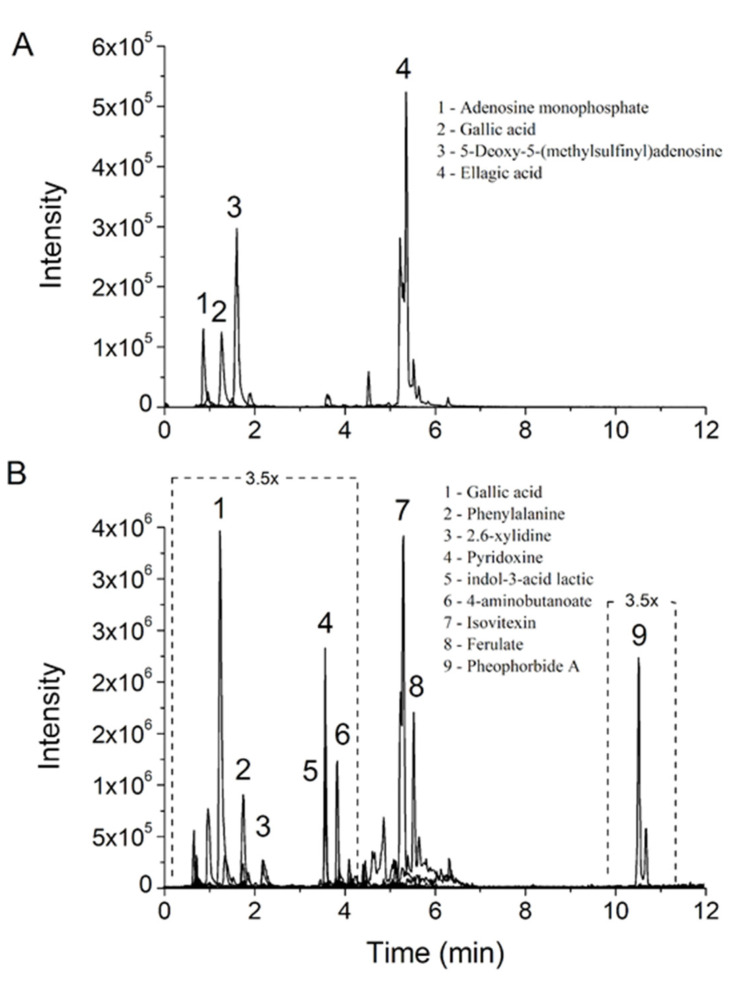
LC-MS/MS fingerprint of *L. rigida* extracts: (**A**) *L. rigida* aqueous leaf extract; (**B**) *L. rigida* hydroethanolic leaf extract. 3.5 x denotes the magnification applied in the dotted areas of the chromatogram.

**Figure 2 molecules-27-06291-f002:**
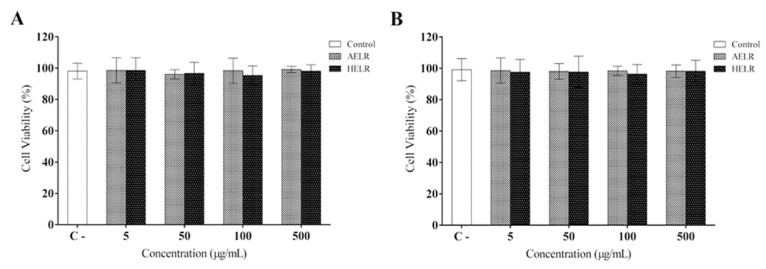
AELR and HELR cytotoxicity effects on RAW 264.7 murine macrophage cells: (**A**) cell viability measured by MTT assay; (**B**) cell viability measured by Alamar Blue^®^ assay. Culture medium DMEM was used as a negative control for cytotoxicity.

**Figure 3 molecules-27-06291-f003:**
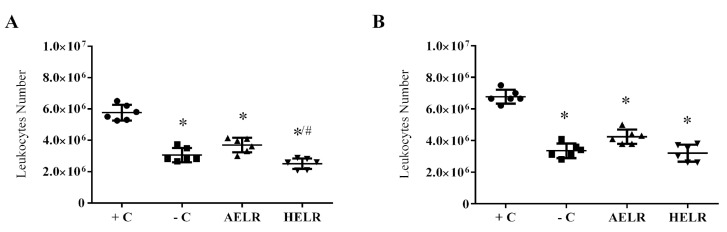
Total leukocyte count in (**A**) LPS-induced peritonitis model and (**B**) differential leukocyte count in an LPS-induced peritonitis model. *L. rigida* aqueous leaf extract (AELR); *L. rigida* hydroethanolic leaf extract (HELR), −C (negative control: animals not induced with LPS) and +C (positive control: animals induced with LPS and treated with PBS). Mice were injected intraperitoneally with 100 µL of LPS 1 h before the intravenous injection of PBS, AELR, and HELR (25 mg/kg/animal). After 4 h, the peritoneal lavage was evaluated for the total number of cells and compared with the LPS-stimulated positive group; * *p* < 0.05; # *p* < 0.05 between the extracts. *n* = 6.

**Figure 4 molecules-27-06291-f004:**
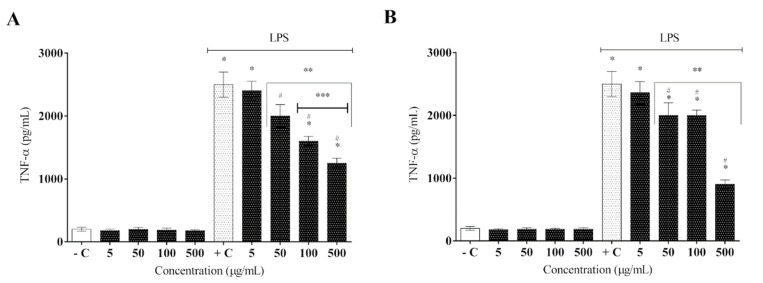
Effect of *L. rigida* extracts on TNF-α cytokine release: (**A**) AELR and (**B**) HELR. The cytokine content was released in RAW 264.7 cells and stimulated by LPS after 24 h. Release of cytokines was quantified by ELISA analysis. Data represent the mean ± SEM from three independent experiments. One-way ANOVA was used followed by a post hoc Tukey test. * *p* < 0.05 vs. the negative control group; # *p* < 0.05 vs. the LPS-stimulated cells; ** *p* < 0.05 between the concentrations of the extract; *** *p* < 0.05 between the higher extract concentrations (100 and 500 µg/mL).

**Figure 5 molecules-27-06291-f005:**
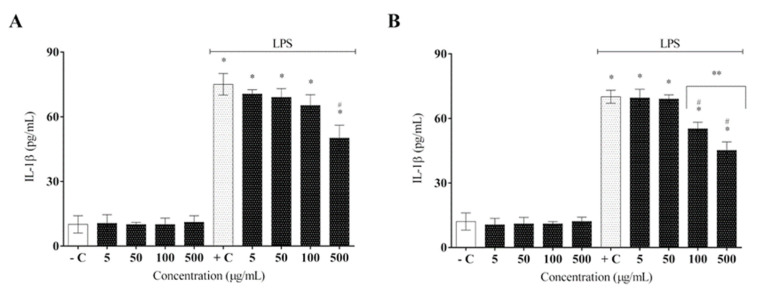
Effect of *L. rigida* extracts on IL-1β cytokine release: (**A**) AELR and (**B**) HELR. The cytokine content was released in RAW 264.7 cells and stimulated by LPS after 24 h. Release of cytokines was quantified by ELISA analysis. Data represent the mean ± SEM from three independent experiments. One-way ANOVA was used followed by a post hoc Tukey test. * *p* < 0.05 vs. the negative control group; # *p* < 0.05 vs. the LPS-stimulated cells; ** *p* < 0.05 between extract concentrations.

**Figure 6 molecules-27-06291-f006:**
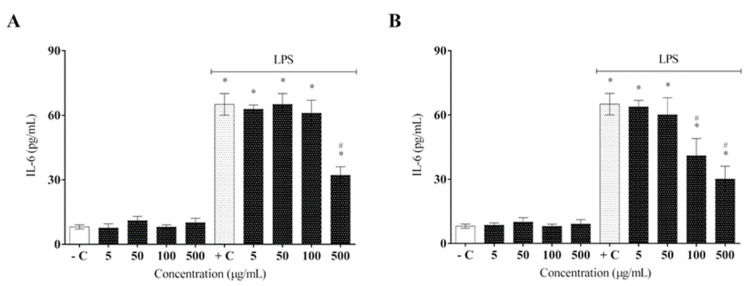
Effect of *L. rigida* extracts on IL-6 cytokine release: (**A**) AELR and (**B**) HELR. The cytokine content was released in RAW 264.7 cells and stimulated by LPS after 24 h. Release of cytokines was quantified by ELISA analysis. Data represent the mean ± SEM from three independent experiments. One-way ANOVA was used followed by a post hoc Tukey test. * *p* < 0.05 vs. the negative control group; # *p* < 0.05 vs. the LPS-stimulated cells.

**Figure 7 molecules-27-06291-f007:**
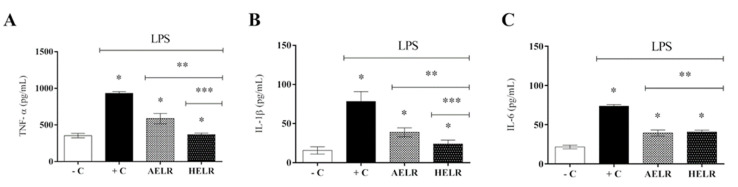
LPS-stimulated cytokine production in C57BL/6 mice. Effect of extracts on: (**A**) TNF-α secretion, (**B**). IL-1β secretion and (**C**) IL-6 secretion. AELR (*L. rigida* aqueous leaf extract). HELR (*L. rigida* hydroethanolic leaf extract). −C (negative control: animals not induced with LPS). +C (positive control: animals induced with LPS and treated with PBS). Results represent mean cytokine levels (pg/mL) as well as standard deviations between animals in each group. Symbols indicate a statistical difference (*p* < 0.05) by ANOVA, with Tukey post-test in the comparison between compounds. * *p* < 0.05 vs. the negative control group; ** *p* < 0.05 vs. the LPS-stimulated group; *** *p*< 0.05 between the extracts. *n* = 6.

**Figure 8 molecules-27-06291-f008:**
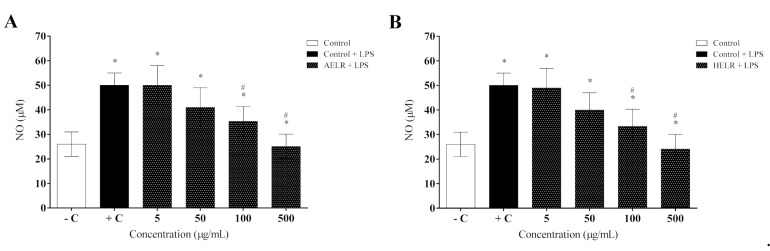
Inhibitory effects of *L rigida* extracts on LPS-stimulated nitric oxide (NO) production in RAW 264.7 macrophages: (**A**) AELR and (**B**) HELR. The NO level in the culture medium was quantified using Griess reagent. Data represent the mean ± SEM from three independent experiments. One-way ANOVA was used followed by a post hoc Tukey test. * *p* < 0.05 vs. the negative control group; # *p* < 0.05 vs. the LPS-stimulated cells.

**Figure 9 molecules-27-06291-f009:**
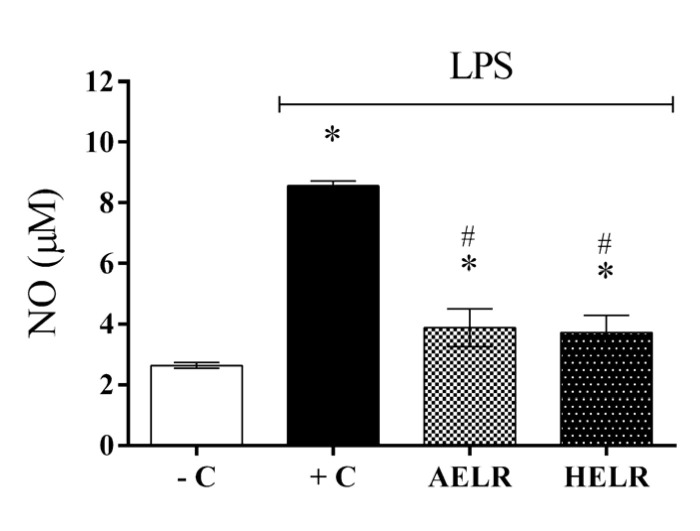
LPS-stimulated nitric oxide production in C57BL/6 mice. AELR (*L. rigida* aqueous leaf extract). HELR (*L. rigida* hydroethanolic leaf extract). –C (negative control: animals not LPS-induced). +C (positive control: animals induced with LPS and treated with PBS). Results represent the means of NO (µM) levels, as well as standard deviations between animals in each group. Symbols indicate a statistical difference (*p* < 0.05) by ANOVA, with Tukey post-test in the comparison between compounds. * *p* < 0.05 vs. the negative control group; # *p* < 0.05 vs. the LPS-stimulated group. *n* = 6.

**Table 1 molecules-27-06291-t001:** Phytocomponents identified in *L. rigida* leaf extracts by LC-MS/MS analyses. *L. rigida* aqueous leaf extract (AELR); *L. rigida* hydroethanolic leaf extract (HELR).

Peak	Compound	Cosine	MassDiff	Mass	Molecular Formula	Ion Fragments	Adduct	Extract
1	Adenosine monophosphate	0.89	0.001	348.071	C_10_H_14_N_5_O_7_P	250.09, 178.07, 136.06	[M + H]^+^	AELR
2	Gallic acid	0.97	0	171.029	C_7_H_6_O_5_	153.02, 135.01, 127.04, 125.02, 109.03, 107.01, 81.04	[M + H]^+^	AELR
3	5′-Deoxy-5′-(methylsulfinyl) adenosine	0.91	0.003	314.093	C_11_H_15_N_5_O_4_S	296.08, 164.06, 136.06, 97.03	[M + H]^+^	AELR
4	Ellagic acid	0.86	0.038	303.048	C_14_H_6_O_8_	285.00, 275.02, 257.01, 201.02	[M + H]^+^	AELR
1	Gallic acid	0.94	0	171.029	C_7_H_6_O_5_	153.02, 127.04, 125.02, 109.03, 107.01, 81.04	[M + H]^+^	HELR
2	Phenylalanine	0.93	0	166.086	C_9_H_11_NO_2_	149.06, 120.08, 103.05	[M + H]^+^	HELR
3	2,6-Xylidine	0.96	0	122.096	C_8_H_11_N	105.07, 79.06, 77.04, 51.04	[M + H]^+^	HELR
4	Pyridoxine	0.87	0	170.081	C_8_H_11_NO_3_	152.07, 142.05, 134.06, 124.08, 96.05	[M + H]^+^	HELR
5	DL-Indole-3-lactic acid	0.98	0.001	188.071	C_11_H_11_NO_3_	146.06, 118.07, 65.04	[M + H-H_2_O]^+^	HELR
6	4-Aminobutanoate	0.87	0	104.071	C_4_H_9_NO_2_	87.05, 86.06, 69.04	[M + H]^+^	HELR
7	Isovitexin	0.97	0.003	433.113	C_21_H_20_O_10_	415.10, 313.07, 283.06, 217.05	[M + H]^+^	HELR
8	Ferulate	0.93	0	177.054	C_10_H_10_O_4_	149.06, 145.03, 117.07, 89.04, 63.03	[M + H-H_2_O]^+^	HELR
9	Pheophorbide A	0.89	0.032	593.237	C_35_H_36_N_4_O_5_	533.25, 461.23, 460.22, 433.24	[M + H]^+^	HELR

## Data Availability

Not apllicable.

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
