# Peer review of "Mass Spectrometric Identification of Licania rigida Benth Leaf Extracts and Evaluation of Their Therapeutic Effects on Lipopolysaccharide-Induced Inflammatory Response"

_molecules, 2022, doi:10.3390/molecules27196291_

Round 1

Reviewer 1 Report

Abstract. Aim/Background is an integral part of this section, write about it 

Abstract. Results in the quantitative form shall be placed 

Section 4. Water (1:10 w/v), what is this ratio?

The studies by Linus et al. & Venancio et al. and so on, these studies shall be shifted to a discussion  

The toxicity study was not performed on aqueous and hydroalcoholic extracts, while it is reported in the literature about the toxicity of leaf extract of the plant under investigation, but the reported extract is ethanolic.  

Author Response

Authors' responses to the Editor and Reviewer

General responses

The authors are grateful for the Reviewers' comments regarding the manuscript, which contribute to improving the text. We verified and agreed with the Reviewers' comments and the manuscript was carefully reviewed based on each of the requested comments and suggestions. Thereby, the requested adjustments were made point-by-point to respond to each reviewer, as described below.

Responses to Reviewer 1

  1. Aim/Background is an integral part of this section, write about it.

We agree with the Reviewer's comment that the aim is an integral part of the abstract. As requested by the Reviewer, the text was rewritten, inserting this part. Please see the text red highlighted in the abstract of the revised manuscript.

  1. Results in the quantitative form shall be placed.

As requested by the Reviewer, the results were placed quantitatively. Please see the text red highlighted in the abstract of the revised manuscript.

  1. Section 4. Water (1:10 w/v), what is this ratio?

A typo. The sentence has been corrected and the 1:10 p/v ratio eliminated. The volume of solvent used in the decoction procedure was specified. Please see item "4.1. Collection of plant material and preparation of extracts" in the revised manuscript.

  1. The studies by Linus et al. & Venancio et al. and so on, these studies shall be shifted to a discussion.

We agree with the Reviewer's suggestion. Thus, texts referring to studies by Linus et al. & Venancio et al. shifted form to the discussion. Please see the red highlighted text of the discussion on page 10 of 17 in the revised manuscript.

  1. The toxicity study was not performed on aqueous and hydroalcoholic extracts, while it is reported in the literature about the toxicity of leaf extract of the plant under investigation, but the reported extract is ethanolic.

As requested by the Reviewer, we clarified in the text why not perform toxicity tests with aqueous and hydroalcoholic extracts. These assays are part of previously published studies by our group using aqueous and hydroalcoholic extracts with a focus on other aspects. Please see the red highlighted text of the discussion on page 10 of 17 in the revised manuscript.

Reviewer 2 Report

I am satisfied with the present version of the manuscript

Author Response

Authors' responses to the Editor and Reviewer

General responses

The authors are grateful for the Reviewers' comments regarding the manuscript, which contribute to improving the text. We verified and agreed with the Reviewers' comments and the manuscript was carefully reviewed based on each of the requested comments and suggestions. Thereby, the requested adjustments were made point-by-point to respond to each reviewer, as described below.

Responses to Reviewer 2

I am satisfied with the present version of the manuscript

We, the authors, sincerely appreciate your comment regarding our manuscript.

Reviewer 3 Report

This study entitled “Mass Spectrometric Identification of Licania rigida Benth Leaf Extracts and Evaluation of Their Therapeutic Effects on Lipopolysaccharide-induced Inflammatory Response” used both in vitro and in vivo models to investigate the anti-inflammatory effects of AELR and HELR. The results showed that both AELR and HELR possess protective effects against LPS-induced inflammatory responses in the RAW 264.7 cells and mice. However, some major points raised shown below

1.     In the abstract, “gallic and ellagic acids as the main constituent in both extracts, while isovitexin, ferulate, some bulky amino acids (e.g., phenylalanine), pheophorbide, lactic acid and pyridoxine were detected in hydroalcoholic extracts”. However, these compounds did not quantified in this study (e.g. using the standard curve to quantify their levels in Licania rigida). Peak area or peak highlight cannot represent their concentrations in the sample. The major bioactive compounds of AELR and HELR should be quantified in this study or it is hard to understand which compound(s) are responsible for the anti-inflammatory effects of AELR and HELR.

2.     This work mentioned that ferulic acid and ellagic acid are the major bioactive compounds of Licania rigida. However, anti-inflammatory effects of ferulic acid and ellagic acid have been reported and comprehensively summarized in review articles (e.g. DOI: 10.1055/a-0633-9492; doi.org/10.1016/j.fct.2017.02.028; DOI: 10.1055/a-0633-9492).

On the basis of the major points mentioned above, this work is lack of novelty and not suitable to be published in Molecules.

Minor points:

1.     In the abstract, “to treat several inflammatory processes”, “ultra-fast liquid chromatography coupled to mass spectrometry” please revise them.

2.     In the abstract, “gallic and ellagic acids” is wrong. Please change to “ gallic acid and ellagic acid”. 

Author Response

Authors' responses to the Editor and Reviewer

General responses

The authors are grateful for the Reviewers' comments regarding the manuscript, which contribute to improving the text. We verified and agreed with the Reviewers' comments and the manuscript was carefully reviewed based on each of the requested comments and suggestions. Thereby, the requested adjustments were made point-by-point to respond to each reviewer, as described below.

Responses to Reviewer 3

This study entitled “Mass Spectrometric Identification of Licania rigida Benth Leaf Extracts and Evaluation of Their Therapeutic Effects on Lipopolysaccharide-induced Inflammatory Response” used both in vitro and in vivo models to investigate the anti-inflammatory effects of AELR and HELR. The results showed that both AELR and HELR possess protective effects against LPS-induced inflammatory responses in the RAW 264.7 cells and mice. However, some major points raised shown below

  • In the abstract, “gallic and ellagic acids as the main constituent in both extracts, while isovitexin, ferulate, some bulky amino acids (e.g., phenylalanine), pheophorbide, lactic acid and pyridoxine were detected in hydroalcoholic extracts”. However, these compounds did not quantified in this study (e.g. using the standard curve to quantify their levels in Licania rigida). Peak area or peak highlight cannot represent their concentrations in the sample. The major bioactive compounds of AELR and HELR should be quantified in this study or it is hard to understand which compound(s) are responsible for the anti-inflammatory effects of AELR and HELR.

We know that phytochemical quantification would increase the interest and impact of the study, and this will be done in the future. This is already described on page 10 of 17 of the revised manuscript. Our analysis was qualitative. Based on this, we modified the text as highlighted in the revised manuscript abstract. Regarding the bioactive compounds of AELR and HELR, we agree that their quantification is important to determine the compounds responsible for the anti-inflammatory effects. Nevertheless, the non-quantification of these compounds does not invalidate the present study, since herbal extracts have been reported as anti-inflammatory agents, probably due to the synergistic action among their components. The use of extracts is the focus of our study and this point is described in the discussion of the manuscript, considering the popular use of extracts to treat diseases associated with inflammatory processes.

  • This work mentioned that ferulic acid and ellagic acid are the major bioactive compounds of Licania rigida. However, anti-inflammatory effects of ferulic acid and ellagic acid have been reported and comprehensively summarized in review articles (e.g. DOI: 10.1055/a-0633-9492; doi.org/10.1016/j.fct.2017.02.028; DOI: 10.1055/a-0633-9492).

As indicated in item 1, an extract qualitative chromatographic analysis was performed, and the peak area or peak highlight cannot represent their concentrations of compounds in the sample, only an indication of their presence in extracts. On the other hand, the anti-inflammatory capabilities of herbal extracts are reported to be a result of the possible synergistic effect of their various components. Thus, ferulic acid and ellagic acid must be placed in the context of this anti-inflammatory action. Furthermore, considering the relevance of the review articles (DOI: 10.1055/a-0633-9492; doi.org/10.1016/j.fct.2017.02.028), these references were accepted and introduced in the revised manuscript. Please see references 64 and 65.

  • On the basis of the major points mentioned above, this work is lack of novelty and not suitable to be published in Molecules.

This work is not lacking in novelty since the experimental approach used is more comprehensive compared to other studies with Licania and other plants. Furthermore, several studies have been published in Molecules showing the importance of evaluating extracts with anti-inflammatory properties aiming at their potential application in the development of herbal formulations.

Minor points

  1. In the abstract, “to treat several inflammatory processes”, “ultra-fast liquid chromatography coupled to mass spectrometry” please revise them.

These sentences have been revised as requested by the Reviewer. Please see the text red highlighted in the revised manuscript.

  1. In the abstract, “gallic and ellagic acids” is wrong. Please change to “gallic acid and ellagic acid”.

We agreed with the Reviewer and changed it to “gallic acid and ellagic acid”. The change is red highlighted in the abstract of the revised manuscript.

Reviewer 4 Report

Evellyn et al. Explored the composition and the anti-inflammatory properties (using lipopolysaccharide stimulation as an inducer of the inflammatory process) of Licania rigida. The paper is within the scope of the journal and the topic is interesting and relevant for the field. The research is well described, and a minor revision of the text editing is necessary. Furthermore, tables and figures are appropriate. In addition, the manuscript is well organized, readily understandable and the research of information has been accurate, with updated references. My suggestion is to investigate the potentiality of Licania rigida to prevent ROS damages.

Author Response

Authors' responses to the Editor and Reviewer

General responses

The authors are grateful for the Reviewers' comments regarding the manuscript, which contribute to improving the text. We verified and agreed with the Reviewers' comments and the manuscript was carefully reviewed based on each of the requested comments and suggestions. Thereby, the requested adjustments were made point-by-point to respond to each reviewer, as described below.

Responses to Reviewer 4

  1. Evellyn et al. Explored the composition and the anti-inflammatory properties (using lipopolysaccharide stimulation as an inducer of the inflammatory process) of Licania rigida.

We appreciate the comment regarding the reference by Evellyn et al. However, we do not understand whether this comment is a statement or a question. If the comment is a statement, our work described in the present manuscript evaluated the anti-inflammatory potential of L. rigida extracts, using LPS as an inflammatory inducer. However, we could not find any references to studies by Evellyn et al. about L. rigida associated with anti-inflammatory properties, and stimulation of lipopolysaccharides as an inducer of the inflammatory process.

  1. The paper is within the scope of the journal and the topic is interesting and relevant for the field.The research is well described, and a minor revision of the text editing is necessary. Furthermore, tables and figures are appropriate. In addition, the manuscript is well organized, readily understandable and the research of information has been accurate, with updated references.

We appreciate the Reviewer's positive comments regarding our manuscript. Also, as requested by the Reviewer, the complete manuscript was edited and proofread by an English editing service.

  1. My suggestion is to investigate the potentiality of Licania rigida to prevent ROS damages.

We agree with the Reviewer's suggestion to investigate the potential of Licania rigida in preventing ROS injury. In this context, we introduce a paragraph highlighted in red on page 11 of 17, considering the involvement of oxidative stress in inflammatory processes. Previously published studies by our group were used in the discussion since these articles evidenced the satisfactory antioxidant capacity of both L. rigida aqueous and hydroethanolic leaf extracts.

Round 2

Reviewer 3 Report

All questions have been answered appropriately.